# Spatial Pattern Characteristics and Factors for the Present Status of Rural Settlements in the Lijiang River Basin Based on ArcGIS

**DOI:** 10.3390/ijerph20054124

**Published:** 2023-02-25

**Authors:** Wenjun Zheng, Wentao Cao, Guifang Li, Sijia Zhu, Xianyan Zhang

**Affiliations:** 1College of Tourism & Landscape Architecture, Guilin University of Technology, Guilin 541004, China; 2Department of Science and Technology, Guilin University of Technology, Guilin 541004, China

**Keywords:** rural settlements, spatial pattern, Lijiang River Basin, ArcGIS, karst landforms

## Abstract

In China, rural settlements have undergone significant changes in response to dramatic socioeconomic shifts. However, there has not been any report on rural settlements in the Lijiang River Basin. In this study, ArcGIS 10.2 (including hot spot analysis and kernel density estimation) and Fragstats 4.2 (such as the landscape pattern index) software were used to analyze the spatial pattern and causes of rural settlements in the Lijiang River Basin. The Lijiang River Basin is mainly dominated by micro- and small-sized rural settlements with small areas. Moreover, the results of a hot spot analysis showed that micro- and small-sized rural settlements were mainly located in the upper reaches, and medium- and large-sized rural settlements were mainly located in the middle and lower reaches. The kernel density estimation results showed that the distribution characteristics of the rural settlements in the upper, middle, and lower reaches were significantly different. The spatial forms of rural settlements were affected by physiographic factors such as elevation and slope, karst landforms, and river trunk channels as well as the national policy system, tourism economic development, town distribution, historical heritage, and minority culture. This study is the first to systematically elaborate on the rural settlement pattern and its internal logic from the perspective of the Lijiang River Basin, providing a basis for the optimization and construction of the rural settlement pattern.

## 1. Introduction

Rural settlements are dominated by agricultural production activities, and their pattern distribution characteristics imply the understanding of the natural environment of the villagers as well as important information such as the degree of civilization, social patterns, and changes in regional characteristics [1]. Studying the spatial evolution of rural settlements can provide insight into the relationship between rural people and the land and contribute to the development of rural geography [2]. For example, by analyzing the changes in rural settlements under the background of tourism, settlements used for tourism accommodation can be more reasonably planned [3]. In addition, optimizing the spatial distribution of rural settlements based on the “quality of life theory” is conducive to improving the quality of life of residents, saving land, and promoting sustainable development [4]. A study found that the rural settlements in Tongzhou District had four main evolution modes in the past 50 years: extinction, diffusion, filling, and merging. The results provide an important reference for the redistribution of local land use patterns [5]. Moreover, Fleisher investigated the changes in rural settlements in Swahili towns between A.D. 750 and 1500 and found that the reunions of scattered rural settlements not only created cohesive new communities but also contributed to economic development [6]. These studies show that analyzing the spatial distribution of rural settlements has guiding significance for optimizing land use efficiency and economic development.

In China, due to the urban–rural conflict of rapid urbanization and the importance of national policies on rural settlements, the spatial types, distribution patterns, and evolutionary characteristics of rural settlements have become a hot topic of academic interest. Studies by foreign scholars on rural settlements include the evolution of settlement structure [7]; the relationship between population, industry, and settlement [8]; sustainable development [9]; and development assessment models [10]. In China, scholars have mainly studied the rural population [11], spatial and temporal evolution trends [5], ecological disturbance [12], the optimal adjustment of patterns [4], etc. However, few scholars have systematically deciphered the inner logic of rural settlement pattern formation from the perspective of the river basin.

In this study, ArcGIS 10.2 software technology, combined with the landscape pattern index calculated by Fragstats 4.2, were used to explore the distribution characteristics of rural settlements in the Lijiang River Basin. The driving factors affecting the spatial characteristics of rural settlements were analyzed from the perspectives of physical geography, social economy, and history. The purpose of this study was to explore the spatial pattern of rural settlements in the Lijiang River Basin and conduct a rudimentary analysis of the reasons for the formation of the current pattern.

## 2. Research Methods and Data Sources

### 2.1. Geographic Area Data

The Lijiang River Basin (110°04′ E~110°45′ E, 24°38′ N~25°53′ N) is located in the central area of Guilin City in the northeast of the Guangxi Zhuang Autonomous Region. The Lijiang River Basin mainly covers six districts, including Deicai District, Xiufeng District, Qixing District, Xiangshan District, Yanshan District, and Lingui District, and six counties, including Resource County, Xing’an County, Lingchuan County, Yangshuo County, Yongfu County, and Pingle County. The topography of the Lijiang River Basin is high in the north and low in the south, while it is high in the east and west and low in the middle, with an overall zonal distribution. The area upstream of the Lijiang River Basin is covered with forests, while the middle and lower downstream sections flow through hilly and karst landform areas with typical karst landform characteristics. The watershed area of the Lijiang River is 6412.72 km^2^, and the total length of the main stream is 164 km (Figure 1).

To describe the distribution of rural settlements in different watershed segments of the Lijiang River Basin, it was divided into three watershed segments, the Xing’an-Lingqu segment, the two rivers and four lakes segment, and the Yangshuo-Yulong River segment (Figure 2), taking into account county boundaries. For easy description, they are abbreviated as the upstream section, the midstream section, and the downstream section, respectively.

### 2.2. Data Sources

Data sources for this study: (1) Remote-sensing monitoring data of land utilization status in China in 2020 (resolution: 30 m) were classified at the secondary level and divided into 23 categories according to the natural attributes of land resources. The data mainly included the distribution ranges of rural settlements, the ranges of rivers and lakes, etc. Rural residential locations were used to characterize “rural settlements”, traditional villages were selected for validation, and the sporadic patch of rural residential locations below 3000 m^2^ was excluded. (2) A vector map of county units in the Guangxi Zhuang Autonomous Region identified the Lijiang River Basin and the counties where the villages are located. (3) DEM elevation data (resolution: 30 m) of Guilin City, Guangxi Zhuang Autonomous Region, were used to analyze the slope, slope direction, and basin extent of the Lijiang River Basin. (4) Vector data of lithologic landforms in the Guangxi Zhuang Autonomous Region were obtained from ISRIC [13]. The scale was 1:1 million.

### 2.3. Getis-Ord General G Analyses

The clustering in the density distribution of rural settlements was assessed using the Getis-Ord General G statistic in ArcGIS v. 10.2 software. The formula is as follows:$$G\left(d\right)=\overset{n}{\underset{i=1}{\sum }}\overset{n}{\underset{j=1}{\sum }}W_{ij}\left(d\right)x_{i}x_{j}/\overset{n}{\underset{i=1}{\sum }}\overset{n}{\underset{j=1}{\sum }}x_{i}x_{j},\;\;\;\forall j\neq i$$
where d denotes the distance; n denotes the number of features in the dataset; W_ij_ (d) denotes the spatial weights between attributes i and j defined by the distance rule; x_i_ and x_j_, respectively, denote the attribute values of i and j; and ∀j ≠ i means that i and j cannot be the same attribute value. When the z-value is scored as positive and the General G observation is greater than the General G expectation, it indicates a high-value clustering of attributes in the study area; otherwise, it is a low-value clustering.

### 2.4. Hot Spot Analysis (Getis-Ord Gi*)

A hot spot analysis was utilized to explore the spatial cluster arrangements appearing in the data. It was performed using the Getis-Ord Gi* statistic to identify hot spots (high values) and cold spots (low values) for rural settlements in ArcGIS v. 10.2 software based on the rural settlements of the Lijiang River Basin in 2020. The formula is as follows:$$Gi^{*}\left(d\right)=\overset{n}{\underset{j=1}{\sum }}w_{ij}\left(d\right)x_{j}\;/\;\overset{n}{\underset{j=1}{\sum }}x_{j}$$
where d denotes the distance; n denotes the number of features in the dataset; W_ij_ (d) denotes the spatial weights between attributes i and j defined by the distance rule; and x_j_ denotes the attribute value of j. The Z-value was obtained from this formula and then further processed as follows:$$Z\left(Gi*\right)=\frac{Gi*-E\left(Gi*\right)}{var\;\left(Gi*\right)}$$

If the Z-value is positive and the score is higher, the hot spots (high values) are more tightly clustered; if the Z-value is negative and the score is lower, the cold spots (low values) are more tightly clustered.

### 2.5. Kernel Density Estimation

Kernel density estimation is utilized to predict the unknown density functions in probability theory, which can directly reflect the distribution characteristics of sample data. The central tendency and dispersion degree of rural settlements were assessed via kernel density estimation [14] using the following formula:$$f\left(x,y\right)=\frac{1}{nh^{2}}\overset{n}{\underset{i=1}{\sum }}k\left(\frac{d_{i}}{n}\right)$$
where f (x, y) is the density value at the location (x, y) of the raster cell center point; n denotes the observed value in the study range; h > 0 denotes the bandwidth, window, or smoothing parameter; k(.) is the kernel function; and d_i_ denotes the distance from the i-th observation point to point (x, y).

### 2.6. Landscape Pattern Index

The landscape pattern index is a method of describing landscape pattern characteristics using indexed landscape features and a quantitative indicator reflecting the relationship between the landscape pattern structure and phenomenological processes. Here, the landscape pattern index was applied to the analysis of the relationship of structural features to the spatial distribution in rural settlement landscapes. Fragstats software was used to analyze the landscape pattern characteristics of rural settlements in the Lijiang River Basin based on the number of patches (NP), the patch density (PD), the patch type area (CA), the proportion of landscape area occupied by patches (PLAND), the average patch size (AREA-MN), the maximum patch index (LPI), the shape index (LSI), the and aggregation index (AI). The specific meanings of the indicators are shown in Table 1.

## 3. Results

### 3.1. The Lijiang River Basin Shows a Micro/Small-Scale and Medium/Large-Scale Rural Settlement Cluster Distribution

In total, 1463 rural settlements in the Lijiang River Basin were identified, with a total area of 11,067.95 hm^2^. The average size of the settlements was 7.56 hm^2^ (Table 2). There were 1070 rural settlements of smaller than average size, accounting for 73% of the total number of settlements. As shown in Table 2, there were 1098 micro/small-scale rural settlements, accounting for 75.1% of the overall number, but their area only accounted for 39.9% of the total area of rural settlements in the entire Lijiang River Basin. However, there were 365 medium/large-scale rural settlements, accounting for 24.9% of the overall number and 50.1% of the overall area. Therefore, the Lijiang River Basin was dominated by micro and small rural settlements with small footprints, and a small number of medium/large rural settlements were distributed within a large area.

The Getis-Ord General G analyses of rural settlements in the Lijiang River Basin showed that the General G observation value (0.000169) was higher than the General G expectation value (0.000088) and the z-score was 1.79 (*p* = 0.0733), indicating that the probability of randomly generating this high-value clustering pattern was less than 10%. Thus, the scale distribution of rural settlements in the Lijiang River Basin was high-value clustering. Therefore, the Lijiang River Basin was affected by the distribution of medium and large rural settlements.

### 3.2. Micro/Small-Scale Villages Are Mainly Located in the Upstream Basin, and Large Villages Are Mainly Located in the Middle and Downstream Basins

The results of the hot spot analysis showed that the red-yellow patches were mainly distributed in the upper reaches of the Lijiang River Basin, with a sparse but global distribution in the tributaries in the middle and lower reaches (Figure 3). Combined with the Z-value, it can be seen that micro/small-scale rural settlements were mainly distributed in the upper reaches of the Lijiang River Basin. For example, as shown in Appendix A, the rural settlements located in the upper section were small and scattered under the influence of the mountains and rivers, which form a typical terraced landscape. Moreover, dark blue patches were observed in the middle and downstream of the Lijiang River Basin and were mainly concentrated near the main stream (Figure 3). Combined with the Z-values, it can be seen that medium and large rural settlements were mainly clustered in the middle and lower reaches of the Lijiang River Basin. For example, as we can see from Appendix A, the rural settlements near the Huixian Wetland in the middle reaches had a characteristic “glass field” (irrigated farmland similar to pieces of glass under the light and shadow), which benefited from the flat terrain of the area, so the rural settlements were large and concentrated. Appendix A shows the rural settlement near Xianggong Hill, located in the downstream section. Moreover, for the characteristic houses of the ethnic minorities in the Guilin area “Stilted Building”, see Appendix A.

In summary, the distribution pattern of rural settlements in the Lijiang River basin was as follows: micro/small-scale and medium/large rural settlements were mainly located in the upstream basin and middle/downstream basins, respectively.

### 3.3. The Rural Settlements in the Upstream, Midstream, and Downstream basins of the Lijiang River Basin, Respectively, Have Banded, Radiating, and Multi-Core Fragmented Distribution

We further interpreted the situation of rural settlements in the Lijiang River Basin from the perspective of a kernel density estimation. As shown in Figure 4, in the upper reaches, rural settlements showed a clear band distribution, but there were still a few rural settlements scattered near the tributaries (green aggregation area). In addition, due to the large span of the midstream basin, the distribution of rural settlements was more numerous and the overall distribution of morphological characteristics was more complex than that of the upstream basin. As can be seen in Figure 4, the distribution of rural settlements followed the direction of river flow in a radial pattern, and there were more red core clusters scattered around without uniform directionality, which were more fragmented. In the lower reaches, the distribution of the red core aggregation areas was fragmented and broken on the whole, with the characteristics a of multi-core distribution, but they were mainly distributed along the main canal and tributaries of the Lijiang River, presenting the scene of the river stringing the core gathering area (Figure 4). Taken together, the distributions of rural settlements in the upper, medium, and lower reaches of the Lijiang River Basin were banded, radiating, and multi-core fragmented, respectively.

### 3.4. The Distribution of Rural Settlements in the Lijiang River Basin from the Perspective of the Landscape Pattern Index

The quantification of landscape patterns through rural settlement metrics is an effective way to analyze the distribution of rural settlements (Table 3). The results showed that the total area and the percentage of rural settlements were the largest in the upper reaches, followed by the medium reaches, and were the smallest in the lower reaches, according to the CA and PLAND. The degree of fragmentation of rural settlements was depicted by NP and PD, showing that the midstream basin was the most fragmented, followed by the downstream basin. The most closely distributed basin was the upstream basin. The combined results of LPI and AREA_MN indicated that medium/large-sized rural settlements occupied more area in the lower reaches and micro/small-sized rural settlements occupied more area in the upstream basin. According to the LSI index, the complexity of the rural settlement shape was greatest in the midstream basin, followed by the upper reaches, and the simplest and most regular shape of rural settlements was observed in the lower reaches. Taken together, in the Lijiang River Basin, the upper reaches were mainly composed of micro/small rural settlements with irregular shapes and a low degree of fragmentation; in the middle reaches, medium/large rural settlements were mainly clustered, irregular in shape, and highly fragmented; and in the downstream basin, medium/large rural settlements had the highest degree of aggregation, a regular shape, and a high degree of fragmentation. 

### 3.5. Elevation and Slope Affect the Spatial Form of Rural Settlements in the Lijiang River Basin

According to Figure 5 and Appendix A, the rural settlements in the Lijiang River Basin were mainly distributed in the elevation range of 100–200 (1214/1463, 82.98%), with an area of 9588.72 that account for 86.6% of the total area of rural settlements. Moreover, the number and scale of rural settlements tended to rise and then decline with an increase in slope, reaching a peak with a gentle slope (>2°–5°) (Figure 6 and Appendix A). Grossly, rural settlements were distributed within the slope range of 0.5°–15°, which shows that rural settlements preferred flat and open areas with smaller slopes. As can be seen in Figure 6, the slope of the middle reaches is smaller and the overall terrain is flatter, so the villages were more concentrated and larger. The southern end of the lower reaches has a smaller slope on both sides of the river, where medium/large-scale rural settlements were clustered and distributed. Therefore, these results indicated that the spatial form of the rural settlements in the Lijiang River Basin was affected by elevation and slope.

### 3.6. Karst Landforms Affect the Spatial Form of Rural Settlements in the Lijiang River Basin

The basic pattern of karst landforms in the Lijiang River Basin was created by Tertiary geotectonic movement [15]. There are four main types of rocks in the Lijiang River Basin: granite, slate/phyllite, and limestone (Figure 7). The limestone is distributed in the middle and lower reaches of the Lijiang River Basin, forming scenic spots such as Guilin Karst Peak Forest Plain and Yangshuo Karst Peak Valley, from which the reputation of “East or west, Guilin scenery is best.” originates. In Guilin Karst Peak Plain (the hinterland of the middle reaches of the Lijiang River Basin), the rural settlements were large and clustered; in Yangshuo Karst Peak Valley (the lower reaches of the Lijiang River Basin) there were also medium- to large-scale rural settlements. In short, the spatial distribution of rural settlements represented by human activities was influenced by the karst landform.

### 3.7. River Trunk Channels Affect the Spatial Form of Rural Settlements in the Lijiang River Basin

The Lijiang River was classified into five classes using the river network classification function of ArcGIS v. 10.2. Figure 8 depicts that rural settlements were mainly clustered near first- to third-class rivers with large patch sizes, and small rural settlement patches were scattered near small fourth- to fifth-class rivers. The distance from a village to the nearest river was calculated using the proximity analysis tool of ArcGIS v. 10.2. Rural settlements in the range of 0–0.5 km from rivers were the most numerous, and their area accounted for almost half (48.2%) of the total area of rural settlements (Appendix A). Interestingly, it could be seen that the farther the distance from the river, the smaller the number of rural settlements, as “distance” and “number” showed an obvious inverse correlation. The distribution of river trunk canals affected the spatial distribution of rural settlements in the Lijiang River Basin.

### 3.8. Other Factors Affecting the Spatial Form of Rural Settlements in the Lijiang River Basin

It is well known that the formation of rural settlements is the result of a combination of factors. Combining the social and historical objective factors in the Lijiang River Basin, it was found that the formation and distribution of rural settlements were also impacted by factors including the national policy system, tourism economic development, town distribution, historical heritage, and minority culture.

National policy systems can directly or indirectly affect the development and construction of rural settlements. In the early years of China, the “family planning” policy solved the tension between the people and the land. However, since the population is an important driving factor for the development of rural settlements, the restriction of population growth also limited the number and scale of rural settlements. However, in recent years, the policies of “building a beautiful China”, “rural revitalization”, and “poverty alleviation” have had a great impact on the scale, form, and number of rural settlements. Under the background of rapid urbanization, rural settlements have been impacted by rapid economic development, resulting in the alienation of village forms, population outflow, and gradual decline. In fact, with the support of national policies, many traditional villages have found a new way to avoid gradual decline or even disappearance.

Moreover, town distribution and the tourism economy promote the development of the surrounding rural economy and thus attract villages to gather. Tourism development puts higher demands on infrastructure such as transportation and housing expansion, so it has a certain influence on the external form of rural settlements, as shown in Figure 9, where the red patches represent towns/tourist destinations surrounded by numerous rural settlements.

Recorded as early as the Warring States period, the Lijiang River Basin was an important military center for a period of time. In the changing times, population migration, cultural integration, and military changes have affected the original residential structure, clan construction, and social identity of Guilin and exerted a profound influence on the formation and development of rural settlements [16].

## 4. Discussion

The study of rural settlements is a characterization of the geographical and spatial attributes of rural settlements, mainly studying the formation, development, spatial distribution characteristics, spatial morphology, and spatial pattern evolution of rural settlements and their interrelationship with the geographic environment [17]. The spatial characteristics of rural settlements reflect the relationship between human life, production, and the surrounding geographical environment under the conditions of different productivity levels [18]. In this study, the distribution of rural settlements in the Lijiang River Basin was characterized using different strategies, and we mainly discussed the influences of physical geographic factors (elevation and slope, karst landform, and river trunk channels) on the formation of rural settlements in the Lijiang River Basin. Our research provides a theoretical basis for optimizing and protecting rural settlement patterns and promoting rural revitalization.

Hot spot analysis (Getis-Ord Gi*), kernel density estimation, and the landscape pattern index were the main technical tools used in this study to explore the pattern of rural settlements. In fact, they are common and reliable tools for evaluating the clustering of an element in its spatial location. For example, the spatial and temporal patterns of land use/land cover changes and land surface temperature variations in Bengaluru urban district, India, were analyzed using Getis-Ord Gi* statistics [19]. Guerri et al. mapped and evaluated the distribution of thermal summer diurnal hot and cool spots associated with greening, urban surfaces, and city morphology in the area of Florence in Tuscany (Italy) using Getis-Ord Gi* spatial statistics [20]. GIS-based kernel density was used to examine the spatial configuration of possible land use areas through archaeological legacy data to survey landscape dynamics [21] and was also applied to estimate the spatial pattern of road density and its impact on landscape fragmentation [22]. Moreover, the landscape pattern index was used to compare land use/land cover changes and landscape patterns in two areas in China [23] and has also been utilized to analyze the effects of land use/land cover and landscape patterns on seasonal river water quality in small catchments [24]. In summary, these technical tools are effective strategies for analyzing spatial landscape patterns and spatial configuration characteristics, so they reliably reflect the real spatial pattern of rural settlements in the Lijiang River Basin.

In China, land in urban areas is owned by the state, while land in rural and suburban areas is owned by collectives unless it is owned by the state, as stipulated by law. Individuals only enjoy the right to use and the right to profit from the land. Therefore, national policies can greatly affect the formation of rural settlements. For example, for the “Beautiful Rural Construction” project, the country systematically and comprehensively allocates people, money, and goods to rural areas for six aspects, including village planning, village construction, the ecological environment, economic development, public services, and other aspects. This policy has significantly affected the geographical location, population distribution, landscape morphology, and economic development of rural settlements. Therefore, it is obvious that the distribution of rural settlements in the Lijiang River Basin is also affected by national policies. The phenomenon of national policies changing rural settlements is not unique to China. In the Chittagong Hill Tracts of Bangladesh, after the nationalization of forests, with the accelerated reduction in forest area, the intensification of population migration, and the development of settled agriculture, the scale of rural areas has expanded [25]. The development of rural settlements in Argentinian Patagonia is also affected by the local land use policy [26]. In short, the national policy is one of the important factors that led to the current pattern of rural settlements in the Lijiang River Basin. Interestingly, in turn, the spatial pattern landscape of rural settlements is the basis for guiding and formulating land use policies. Therefore, the analysis of the spatial pattern of rural settlements in the Lijiang River Basin has a certain optimization and protection effect on the existing local rural settlement pattern.

Tourism economic development and town distribution are the formation factors of rural settlements in many countries, including China [27], Nepal’s Annapurna region [3], Israel [28], and NP Kopaonik (Serbia) [9]. In addition, it has also become a consensus that historical heritage is a vital factor in rural settlements because, in some cases, it may be able to determine the future economic specialization of the region and its prospects for social and economic development [29,30,31,32]. Moreover, the influence of minority cultures on the development of rural settlements may be unique to China. Guilin has 28 ethnic minorities (all 55 ethnic groups are minorities except the Han nationality) and is the gathering place of ethnic minorities. Due to the excavation of the Lingqu Canal in history, the Han Chinese took advantage of this waterway to occupy some of the favorable geographical positions to build rural settlements, which were larger and more concentrated. Compared to the Han nationality, due to their fear of external things, ethnic minorities tend to live in deep mountains and valleys with higher elevations and greater slopes. To adapt to the disadvantaged geographical environment, a large amount of ethnic ecological wisdom has been generated, such as terraces, stilted buildings, etc., and their rural settlements are smaller and more scattered due to the geographical conditions.

However, this study had some limitations. First, due to the lack of accuracy of the DEM data and land use data, the landscape ecological pattern index of rural settlements had certain errors, which affected the accuracy of the calculation results and led to bias in the analysis of the results. Second, unmanned aerial vehicles and remote sensing with higher accuracy are needed to increase the accuracy and precision of the data and improve the relevance and validity of the analysis results. Third, this study was based on the data of rural settlements in the Lijiang River Basin in 2020, with a single time node, and lacked research on the evolution of settlement characteristics within a long time series, and the overall analysis framework was relatively macro. The next step can be more focused on the distinctive rural settlements in the Lijiang River Basin.

## 5. Conclusions

Rural settlements in the Lijiang River Basin are mainly micro/small settlements with small area proportions, mainly concentrated in the tributaries of the Lijiang River. Medium/large rural settlements account for a large proportion of the area and are concentrated in the flat middle reaches and the main streams of the lower reaches. The distribution characteristics of different river basins differ significantly. The rural settlements in the upper reaches of the river are zoned along the river, with many small settlements clustered in irregular shapes and a high degree of fragmentation. The rural settlements in the middle river section are radially distributed along the river with a fragmented distribution. There are large rural settlements with irregular shapes and high fragmentation. The rural settlements in the lower reaches are distributed along the main channel of the Lijiang River in a multi-core fragmentation, with medium/large villages gathered. The distribution of rural settlements in the Lijiang River Basin is not only affected by physical geographic factors such as elevation and slope, karst landforms, and river trunk channels but also by human factors such as the national policy system, tourism economic development, town distribution, historical heritage, and minority culture.

## Figures and Tables

**Figure 1 ijerph-20-04124-f001:**
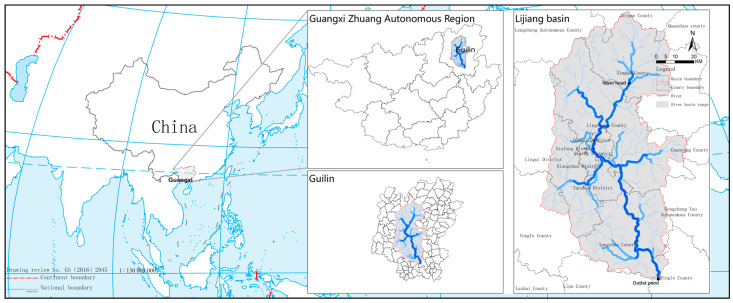
Location of the Lijiang River Basin.

**Figure 2 ijerph-20-04124-f002:**
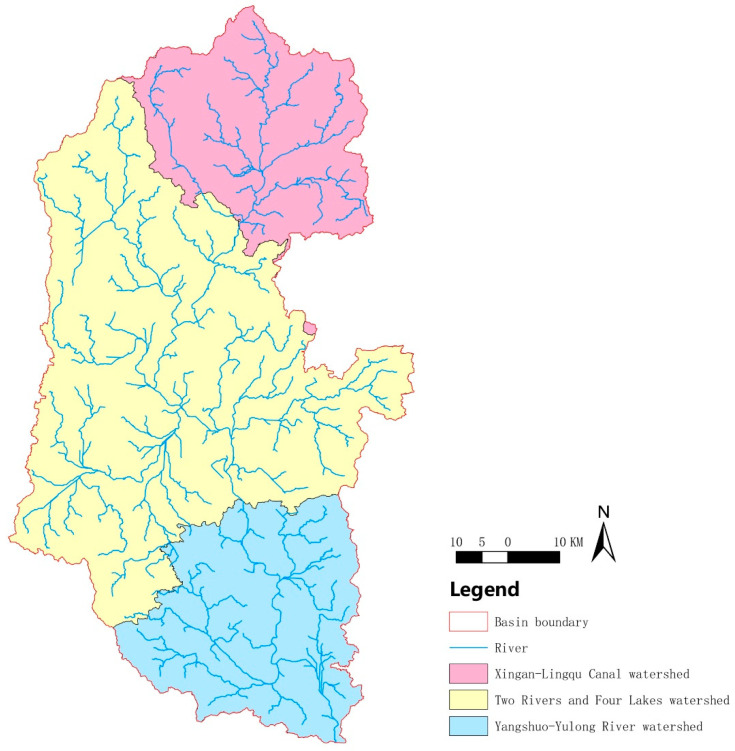
A diagram of the subdivisions of the Lijiang River Basin. The red area is the upstream basin (known locally as the Xing ‘an-Lingqu Canal watershed), the yellow area is the midstream basin (known locally as the two rivers and four lakes watershed), and the blue area is the downstream basin (known locally as the Yangshuo-Yulong River watershed).

**Figure 3 ijerph-20-04124-f003:**
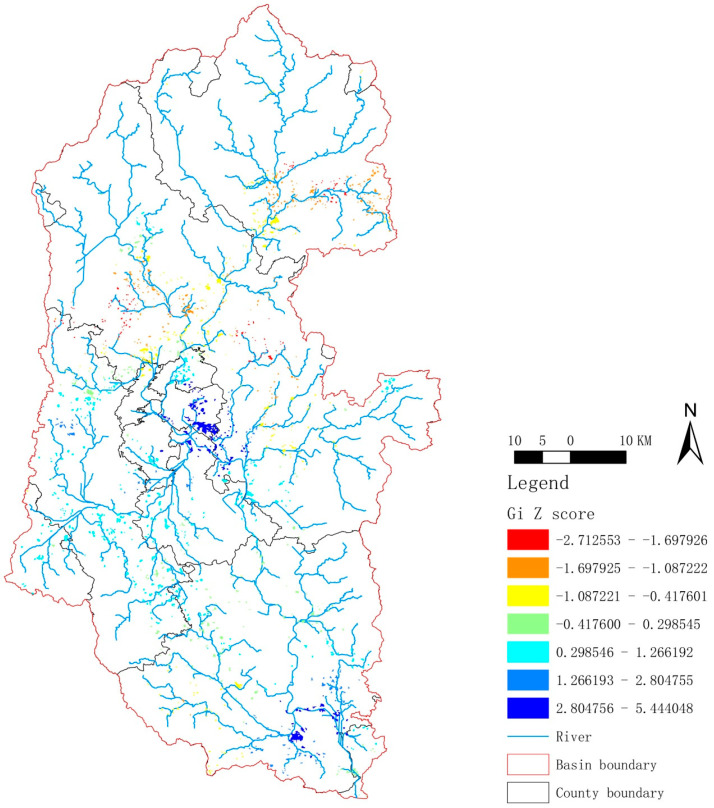
A hot spot plot of rural settlement in the Lijiang River Basin. The colors of the dots from red to blue indicate that the Gi Z-score gradually increased.

**Figure 4 ijerph-20-04124-f004:**
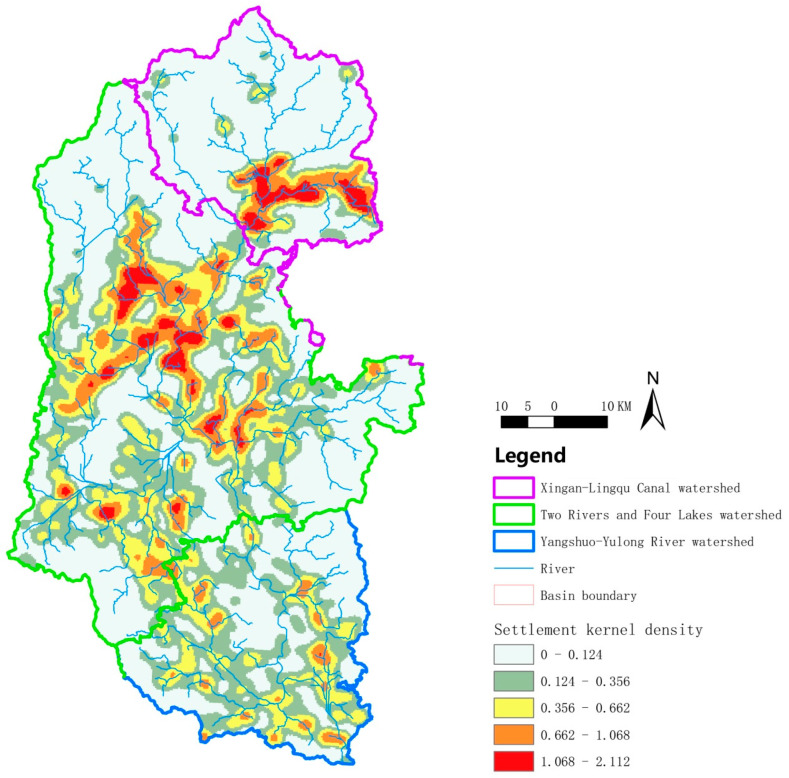
Spatial density distribution plot of rural settlements in the Lijiang River Basin. The redder the patch, the denser the rural settlement.

**Figure 5 ijerph-20-04124-f005:**
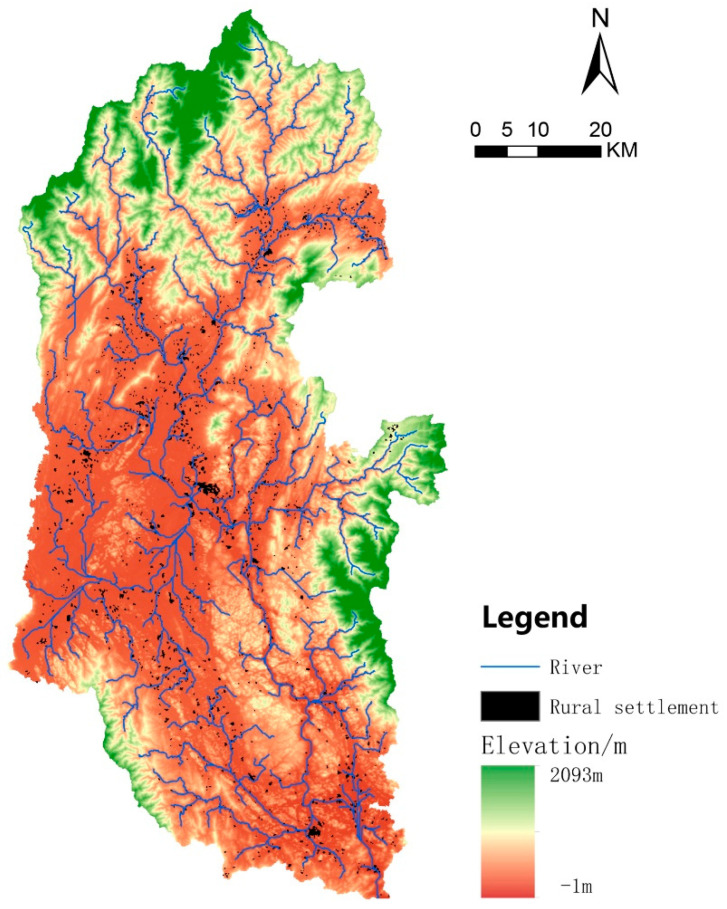
Elevation map of rural settlement distribution in the Lijiang River Basin. Black spots indicate rural settlements.

**Figure 6 ijerph-20-04124-f006:**
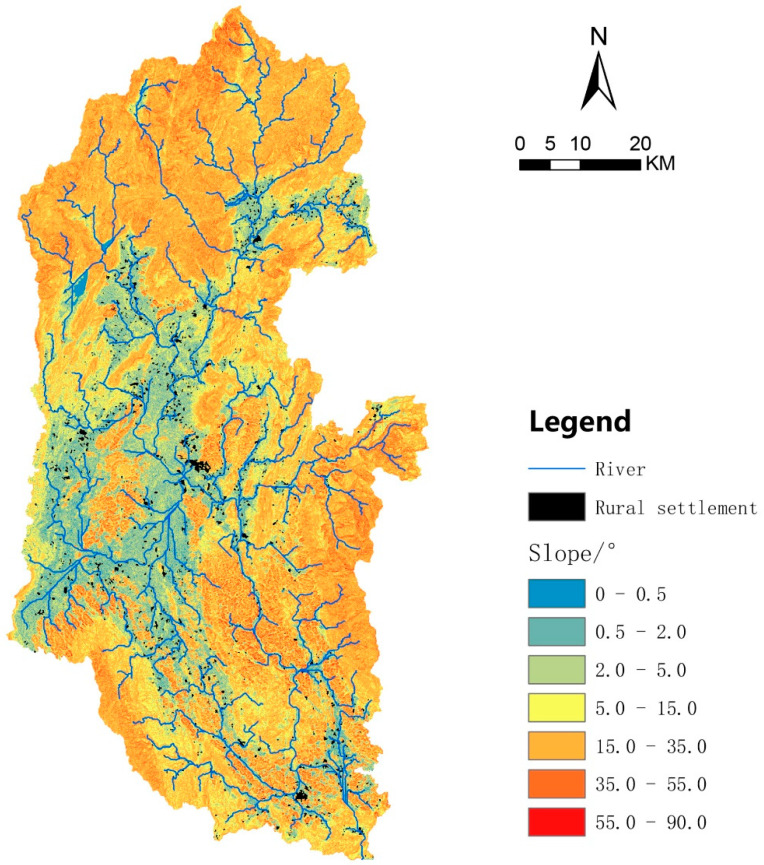
Slope map of rural settlement distribution in the Lijiang River Basin. Black spots indicate rural settlements.

**Figure 7 ijerph-20-04124-f007:**
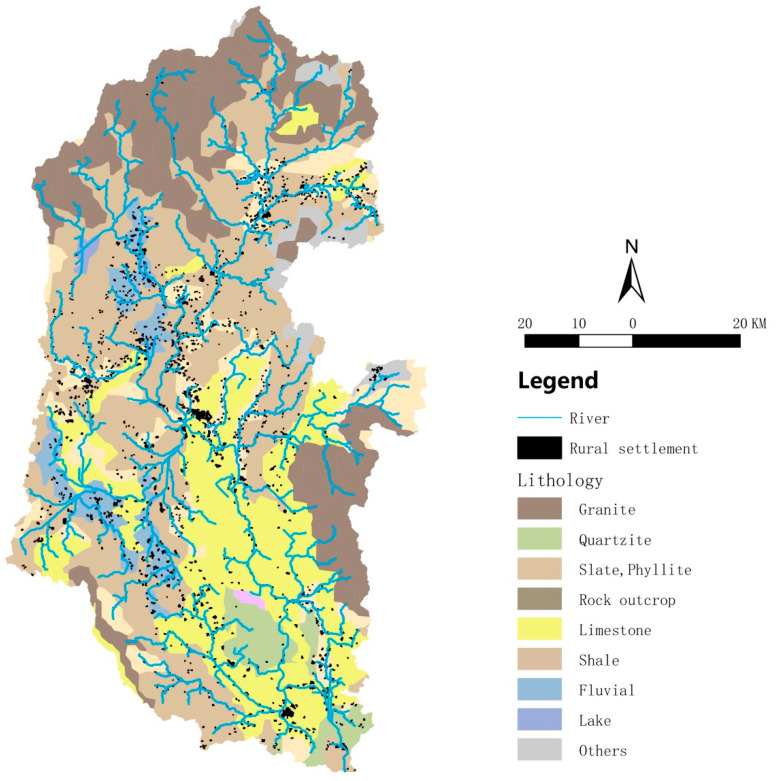
Lithologic geomorphological map of rural settlement distribution in the Lijiang River Basin. Black spots indicate rural settlements.

**Figure 8 ijerph-20-04124-f008:**
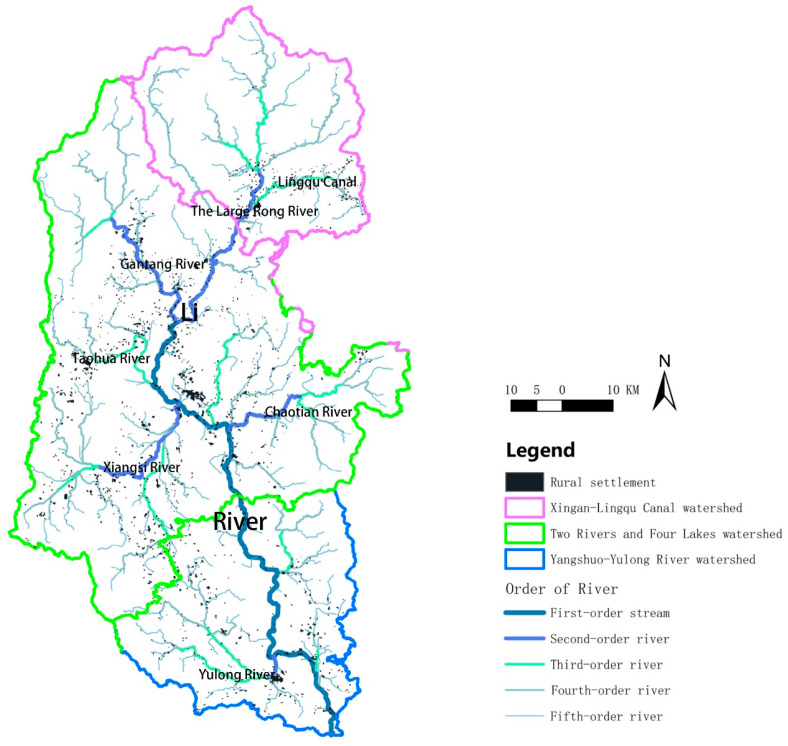
River grading map of rural settlement distribution in the Lijiang River Basin. Black spots indicate rural settlements.

**Figure 9 ijerph-20-04124-f009:**
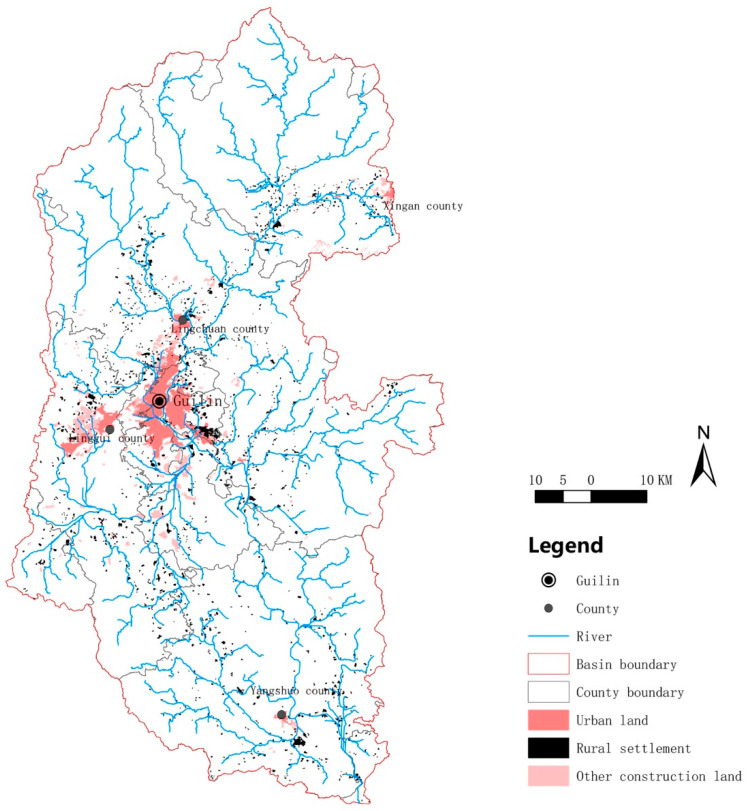
Distribution of urban land and rural settlements in the Lijiang River Basin.

**Table 1 ijerph-20-04124-t001:** The specific meanings of the indicators.

Landscape Pattern Index	Index Type	Indicator Meaning
NP (each)	Poly-dispersion index	The total number of patches. When describing landscape heterogeneity, the value is positively correlated with landscape fragmentation.
PD (each/100 hm^2^)		The number of patches in 100 hm^2^. The value is positively correlated with landscape fragmentation.
CA (hm^2^)	Edge area index	The total area of a certain type of patch.
PLAND (%)		The percentage of a certain type of patch in the overall landscape area.
AREA-MN (hm^2^)		The average area size of a certain type of patch.
LPI (%)		The proportion of the largest patch in a given patch type occupying the entire landscape area.
LSI	Shape index	A larger value indicates a more irregular patch shape.
AI (%)		The larger the value, the greater the aggregation.

**Table 2 ijerph-20-04124-t002:** Statistics of rural settlements of different sizes in the Lijiang River Basin.

Types of Rural Settlements	Micro Type (<2 hm^2^)	Small Type (2–8 hm^2^)	Medium Type (8–20 hm^2^)	Large Type (>20 hm^2^)	Total
Number	128	970	292	73	1463
Area (hm^2^)	209.0	4211.9	3502.6	3144.4	11,068.0

**Table 3 ijerph-20-04124-t003:** Landscape pattern index of rural settlements in the Lijiang River Basin.

Reaches	CA	PLAND	NP	PD	LPI	AREA_MN	LSI	AI
Upper	1293.75	1.01	234	0.18	0.086	5.53	2.15	85.30
Medium	7355.16	2.00	934	0.25	0.12	7.87	2.42	87.09
Lower	2542.77	1.81	300	0.21	0.17	8.48	2.01	87.44

## Data Availability

The data presented in this study are available on request from the corresponding authors.

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
