# Peer review of "Spatial Pattern Characteristics and Factors for the Present Status of Rural Settlements in the Lijiang River Basin Based on ArcGIS"

_ijerph, 2023, doi:10.3390/ijerph20054124_

Round 1
Reviewer 1 Report
Some of the comments for the authors are as follows:
· Please avoid sentences declaring “We found… (line 15), We performed…(line 100), We further interpreted…(line 168)”. I suggest changing those sentences into: It was found…, There was preformed…, It was further interpreted… The same comments for the lines 174, 261, 295, 296.
· It can be noticed in the Abstract that the authors mention the national policy system, tourism economic development, town distribution, historical heritage, and minority culture as factors that affect spatial form of rural settlements, but it is not clearly elaborated and discussed through the results and discussion.
· Lines 48 and 49. The study deals more with the theoretical basis of spatial pattern of rural settlements than with the idea of their protection. This sentence needs enhancement and reformulation.
· The results show detailed classification of the pattern of rural settlements, but the results require deeper explanation by providing readers with the examples and relationships between settlements belonging to the same cluster.
· The Discussion section should deliberate further the impact of clustering, hot spot areas, landscape pattern index on the spatial pattern characteristics of the rural settlements. Are there some similar articles pertaining to this topic? Only two articles have been cited in the Discussion. The authors should provide readers with more articles that are related to this topic, and they should deliberate concreate connections between the statistical aspect of results and situation on the field. Factors such as national policy system, tourism economic development, town distribution, historical heritage, and minority culture are poorly deliberated in the Discussion.
· Conclusion should be improved after making changes in the Discussion part. It should be in accordance with the results and discussion. Please, check the grammar in the line 321.
· The authors should offer more articles pertaining to this topic in order to illustrate relationships between previous research and this one. Reference list should contain wider list of articles related to this topic.
Author Response
Dear Reviewer 1:
On behalf of my co-authors, we thank you very much for giving us an opportunity to revise our manuscript, we appreciate editor and reviewers very much for their positive and constructive comments and suggestions on our manuscript entitled "Spatial pattern characteristics and causes of rural settlements in Lijiang River Basin based on ArcGIS ". (ID: ijerph-2152077).
We have studied reviewer’s comments carefully and have made revision which marked in red in the paper. Those comments are all valuable and very helpful for revising and improving our paper, as well as the important guiding significance to our researches. We have studied comments carefully and have made correction which we hope meet with approval. The main corrections in the paper and the responds to the reviewer’s comments are as following:
Comments and Suggestions for Authors
Some of the comments for the authors are as follows:
- Please avoid sentences declaring “We found… (line 15), We performed…(line 100), We further interpreted…(line 168)”. I suggest changing those sentences into: It was found…, There was preformed…, It was further interpreted… The same comments for the lines 174, 261, 295, 296.
Response: Thank you very much for your suggestions. We have revised the entire text to ensure that there are no first-person statements. We asked our native English colleagues to polish our manuscript. We hope that our manuscript can get your approval.
- It can be noticed in the Abstract that the authors mention the national policy system, tourism economic development, town distribution, historical heritage, and minority culture as factors that affect spatial form of rural settlements, but it is not clearly elaborated and discussed through the results and discussion.
Response: Thank you for your suggestions. According to your comments, we have discussed these factors in the Discussion section to better fit the Abstract and Results. As follows:
In China, land in urban areas is owned by the state, while land in rural and suburban areas is owned by collectives unless it is owned by the state as stipulated by law. Individuals only enjoy the right to use and the right to profit from the land. Therefore, national policies can greatly affect the formation of rural settlements. For example, for the "Beautiful Rural Construction" project, the country systematically and comprehensively allocates people, money, and goods to rural areas from six aspects, including village planning, village construction, ecological environment, economic development, public services, and other aspects. This policy has significantly affected the geographical location, population distribution, landscape morphology, and economic development of rural settlements. Therefore, it is obvious that the distribution of rural settlements in the Lijiang River Basin is also affected by national policies. The phenomenon of national policies changing rural settlements is not unique to China. In the Chittagong Hill Tracts of Bangladesh, after the nationalization of forests, with the accelerated reduction of forest area, the intensification of population migration, and the development of settled agriculture, the scale of rural areas has expanded [25]. The development of rural settlements in Argentinian Patagonia is also affected by the local land use policy [26]. In short, national policy is one of the important factors that lead to the current pattern of rural settlements in the Lijiang River Basin. Interestingly, in turn, the spatial pattern landscape of rural settlements is the basis for guiding and formulating land use policies. Therefore, the analysis of the spatial pattern of rural settlements in the Lijiang River Basin has a certain optimization and protection effect on the existing local rural settlement pattern.
Tourism economic development and town distribution are the formation factors of rural settlements in many countries, including China [27], Nepal's Annapurna region [28], Israel [29], and NP Кopaonik (Serbia) [9]. In addition, it has also become a consensus that historical heritage is a vital factor in rural settlements, because in some cases, it may be able to determine the future economic specialization of the region and its prospects for social and economic development [30-33]. Moreover, the influence of minority cultures on the development of rural settlements may be unique to China. Guilin has 28 ethnic minorities (all 55 ethnic groups are minorities except the Han nationality), which is the gathering place of ethnic minorities. Due to the excavation of the Lingqu Canal in history, the Han Chinese took advantage of this waterway to occupy part of the favorable geographical positions to build rural settlements, which are larger and more concentrated. Compared with the Han nationality, due to their fear of external things, ethnic minorities tend to live in deep mountains and valleys with higher elevations and greater slopes. To adapt to the disadvantaged geographical environment, a large number of ethnic ecological wisdom has been generated, such as terraces and stilted buildings, etc., and their rural settlements will be smaller and more scattered due to the geographical conditions.
- Lines 48 and 49. The study deals more with the theoretical basis of spatial pattern of rural settlements than with the idea of their protection. This sentence needs enhancement and reformulation.
Response: Thank you for your suggestions. We have revised it as follows:
The purpose of this study is to explore the spatial pattern of rural settlements in the Lijiang River Basin and make a rudimentary analysis of the reasons for the formation of the current pattern.
- The results show detailed classification of the pattern of rural settlements, but the results require deeper explanation by providing readers with the examples and relationships between settlements belonging to the same cluster.
Response: Thank you for your suggestions. We have listed some examples in the article and added representative images. As follows:
In the section “3.2. Micro/small-scale villages are mainly located in the upstream, and large villages are mainly located in the middle and downstream”
“…As shown in Supplemental Figure 1, the rural settlements located in the upper section are small and scattered under the influence of the mountains and rivers, and form a typical terraced landscape…”
“…For example, as we can see from Supplemental Figure 2, the rural settlements near Huixian Wetland in the middle reaches have characteristic "glass field" (farmland irrigated, like pieces of glass under the light and shadow), which benefits from the flat terrain of the area, so the rural settlements are large and concentrated…”
Supplemental Figure 3 showed the rural settlement near Xianggong Hill, located in the downstream section.
Moreover, the characteristic houses of the ethnic minorities in Guilin area, "Stilted Building" see Supplemental Figure 4.
- The Discussion section should deliberate further the impact of clustering, hot spot areas, landscape pattern index on the spatial pattern characteristics of the rural settlements. Are there some similar articles pertaining to this topic? Only two articles have been cited in the Discussion. The authors should provide readers with more articles that are related to this topic, and they should deliberate concreate connections between the statistical aspect of results and situation on the field. Factors such as national policy system, tourism economic development, town distribution, historical heritage, and minority culture are poorly deliberated in the Discussion.
Response: Thank you for your suggestions. We have done as request, and added more related references in discussion. As follows:
For “The Discussion section should deliberate further the impact of clustering, hot spot areas, landscape pattern index on the spatial pattern characteristics of the rural settlements. Are there some similar articles pertaining to this topic?”, we added the following discussion:
Hot spot analysis (Getis-Ord Gi*), kernel density estimation, and landscape pattern index are the main technical tools used in this study to explore the pattern of rural settlements. In fact, they are a common and reliable tool for evaluating the clustering of an element in its spatial location. For example, the spatial and temporal patterns of land use land cover changes and land surface temperature variations in Bengaluru urban district, India, was analyzed using Getis–Ord Gi* statistics [19]. Guerri et al. mapped and evaluated the distribution of thermal summer diurnal hot- and cool-spots associated with greening, urban surfaces, and city morphology in the area of Florence in Tuscany (Italy) using Getis-Ord Gi* spatial statistics [20]. GIS-based kernel density was used to examine the spatial configuration of possible land-use areas through archaeological legacy data to survey landscape dynamics [21] and also applied to estimate the spatial pattern of road density and its impact on landscape fragmentation [22]. Moreover, the landscape pattern index was used to compare land use/land cover change and landscape pattern in two areas in China [23] and has also been utilized to analyze the effects of land use/land cover and landscape patterns on seasonal river water quality in small catchments [24]. In summary, these technical tools are effective strategies for analyzing spatial landscape patterns and spatial configuration characteristics, so they reliably reflect the real spatial pattern of rural settlements in the Lijiang River Basin.
For “Factors such as national policy system, tourism economic development, town distribution, historical heritage, and minority culture are poorly deliberated in the Discussion.”
In China, land in urban areas is owned by the state, while land in rural and suburban areas is owned by collectives unless it is owned by the state as stipulated by law. Individuals only enjoy the right to use and the right to profit from the land. Therefore, national policies can greatly affect the formation of rural settlements. For example, for the "Beautiful Rural Construction" project, the country systematically and comprehensively allocates people, money, and goods to rural areas from six aspects, including village planning, village construction, ecological environment, economic development, public services, and other aspects. This policy has significantly affected the geographical location, population distribution, landscape morphology, and economic development of rural settlements. Therefore, it is obvious that the distribution of rural settlements in the Lijiang River Basin is also affected by national policies. The phenomenon of national policies changing rural settlements is not unique to China. In the Chittagong Hill Tracts of Bangladesh, after the nationalization of forests, with the accelerated reduction of forest area, the intensification of population migration, and the development of settled agriculture, the scale of rural areas has expanded [25]. The development of rural settlements in Argentinian Patagonia is also affected by the local land use policy [26]. In short, national policy is one of the important factors that lead to the current pattern of rural settlements in the Lijiang River Basin. Interestingly, in turn, the spatial pattern landscape of rural settlements is the basis for guiding and formulating land use policies. Therefore, the analysis of the spatial pattern of rural settlements in the Lijiang River Basin has a certain optimization and protection effect on the existing local rural settlement pattern.
Tourism economic development and town distribution are the formation factors of rural settlements in many countries, including China [27], Nepal's Annapurna region [28], Israel [29], and NP Кopaonik (Serbia) [9]. In addition, it has also become a consensus that historical heritage is a vital factor in rural settlements, because in some cases, it may be able to determine the future economic specialization of the region and its prospects for social and economic development [30-33]. Moreover, the influence of minority cultures on the development of rural settlements may be unique to China. Guilin has 28 ethnic minorities (all 55 ethnic groups are minorities except the Han nationality), which is the gathering place of ethnic minorities. Due to the excavation of the Lingqu Canal in history, the Han Chinese took advantage of this waterway to occupy part of the favorable geographical positions to build rural settlements, which are larger and more concentrated. Compared with the Han nationality, due to their fear of external things, ethnic minorities tend to live in deep mountains and valleys with higher elevations and greater slopes. To adapt to the disadvantaged geographical environment, a large number of ethnic ecological wisdom has been generated, such as terraces and stilted buildings, etc., and their rural settlements will be smaller and more scattered due to the geographical conditions.
- Conclusion should be improved after making changes in the Discussion part. It should be in accordance with the results and discussion. Please, check the grammar in the line 321.
Response: Thank you very much for your suggestion. After making changes in the Discussion part, we have revised the Conclusion, as follows:
Rural settlements in the Lijiang River Basin are mainly micro/small settlements with a small area proportion, mainly concentrated in the tributaries of the Lijiang River. Medium/large rural settlements account for a large proportion of the area and are concentrated in the flat middle reaches and the main streams of the lower reaches. The distribution characteristics of different river basins differ significantly. The rural settlements in the upper reaches of the river are zoned along the river, with many small settlements clustered in irregular shapes and a high degree of fragmentation. The rural settlements in the middle river section are radially distributed along the river and fragmented distribution. There are large rural settlements with irregular shapes and high fragmentation. The rural settlements in the lower reaches are distributed along the main channel of the Lijiang River in a multi-core fragmentation, with medium/large villages gathered. The distribution of rural settlements in the Lijiang River Basin is not only affected by physical geographic factors such as elevation and slope, karst landform, and river trunk channels, but also by human factors such as national policy system, tourism economic development, town distribution, historical heritage, and minority culture.
- The authors should offer more articles pertaining to this topic in order to illustrate relationships between previous research and this one. Reference list should contain wider list of articles related to this topic.
Response: Thank you very much for your suggestion. We have revised the discussion section in response to your previous comments and introduced a number of similar articles. Please review it. If you still have any questions, please do not hesitate to contact us.
We would like to express our great appreciation to you and reviewers for comments on our paper. Looking forward to hearing from you.
Thank you and best regards.
Yours sincerely,
Xianyan Zhang

Reviewer 2 Report
In the title: Spatial pattern characteristics and causes of rural settlements in 2 Lijiang River Basin based on ArcGIS, the word causes seems out of context or not explained enough; causes of what? Which causes?
The specific methodology should be a bit more elaborated in the abstract
The manuscript is well-written and informative. The theoretical background in section 1. Introduction research should be more extensive and include previous similar research studies and in section 4. Discussion the implications of the research results with the previous similar studies should be discussed.
Sections 4. and 5. could be merged into one integrated section.
Author Response
Dear Reviewer 2:
On behalf of my co-authors, we thank you very much for giving us an opportunity to revise our manuscript, we appreciate editor and reviewers very much for their positive and constructive comments and suggestions on our manuscript entitled "Spatial pattern characteristics and causes of rural settlements in Lijiang River Basin based on ArcGIS ". (ID: ijerph-2152077).
We have studied reviewer’s comments carefully and have made revision which marked in red in the paper. Those comments are all valuable and very helpful for revising and improving our paper, as well as the important guiding significance to our researches. We have studied comments carefully and have made correction which we hope meet with approval. The main corrections in the paper and the responds to the reviewer’s comments are as following:
In the title: Spatial pattern characteristics and causes of rural settlements in 2 Lijiang River Basin based on ArcGIS, the word causes seems out of context or not explained enough; causes of what? Which causes?
Response: Thank you very much for your comments. The word causes refers to the factors causing the present situation of rural settlement distribution in the Lijiang River Basin, the “causes” including physiographic factor such as elevation and slope, Karst landforms, and river trunk channels, and human factors such as national policy system, tourism economic development, town distribution, historical heritage, and minority culture. After considering your comments, we changed “causes” to “factors” to avoid readers' confusion. As follows:
Spatial pattern characteristics and factors for the present status of rural settlements in Lijiang River Basin based on ArcGIS
The specific methodology should be a bit more elaborated in the abstract
Response: Thank you very much for your suggestions. We have added details in the methodology in the abstract, as follows:
In this study, ArcGIS (including hot spot analysis and kernel density estimation) and Fragstats (such as landscape pattern index) software were used to analyze the spatial pattern and causes of rural settlements in Lijiang River Basin.
The manuscript is well-written and informative. The theoretical background in section 1. Introduction research should be more extensive and include previous similar research studies and in section 4. Discussion the implications of the research results with the previous similar studies should be discussed.
Response: Thank you very much for your suggestions. We have done as request, as follows:
For “Introduction research should be more extensive and include previous similar research studies and in section”, we have revised it as follows:
Rural settlements are dominated by agricultural production activities, and their pattern distribution characteristics imply the understanding of the natural environment of the villagers and imply important information such as the degree of civilization, social patterns, and changes in regional characteristics [1]. Studying the spatial evolution of rural settlements can provide insight into the relationship between rural people and land and contribute to the development of rural geography [2]. For example, by analyzing the changes in rural settlements under the background of tourism, settlements used for tourism accommodation can be more reasonably planned [3]. In addition, optimizing the spatial distribution of rural settlements based on the "quality of life theory" is conducive to improving the quality of life of residents, saving land, and promoting sustainable development [4]. The study found that the rural settlements in Tongzhou District had four main evolution modes in the past 50 years: extinction, diffusion, filling, and merging. The results provide an important reference for the redistribution of local land use patterns [5]. Moreover, Fleisher found that investigated the changes in rural settlements in Swahili towns between A.D. 750 and 1500, and found that the reunions of scattered rural settlements not only created cohesive new communities but also contributed to economic development [6]. These studies show that analyzing the spatial distribution of rural settlements has guiding significance for optimizing land use efficiency and economic development.
For “Discussion the implications of the research results with the previous similar studies should be discussed”, we have added some references as follows:
Hot spot analysis (Getis-Ord Gi*), kernel density estimation, and landscape pattern index are the main technical tools used in this study to explore the pattern of rural settlements. In fact, they are a common and reliable tool for evaluating the clustering of an element in its spatial location. For example, the spatial and temporal patterns of land use land cover changes and land surface temperature variations in Bengaluru urban district, India, was analyzed using Getis–Ord Gi* statistics [19]. Guerri et al. mapped and evaluated the distribution of thermal summer diurnal hot- and cool-spots associated with greening, urban surfaces, and city morphology in the area of Florence in Tuscany (Italy) using Getis-Ord Gi* spatial statistics [20]. GIS-based kernel density was used to examine the spatial configuration of possible land-use areas through archaeological legacy data to survey landscape dynamics [21] and also applied to estimate the spatial pattern of road density and its impact on landscape fragmentation [22]. Moreover, the landscape pattern index was used to compare land use/land cover change and landscape pattern in two areas in China [23] and has also been utilized to analyze the effects of land use/land cover and landscape patterns on seasonal river water quality in small catchments [24]. In summary, these technical tools are effective strategies for analyzing spatial landscape patterns and spatial configuration characteristics, so they reliably reflect the real spatial pattern of rural settlements in the Lijiang River Basin.
In China, land in urban areas is owned by the state, while land in rural and suburban areas is owned by collectives unless it is owned by the state as stipulated by law. Individuals only enjoy the right to use and the right to profit from the land. Therefore, national policies can greatly affect the formation of rural settlements. For example, for the "Beautiful Rural Construction" project, the country systematically and comprehensively allocates people, money, and goods to rural areas from six aspects, including village planning, village construction, ecological environment, economic development, public services, and other aspects. This policy has significantly affected the geographical location, population distribution, landscape morphology, and economic development of rural settlements. Therefore, it is obvious that the distribution of rural settlements in the Lijiang River Basin is also affected by national policies. The phenomenon of national policies changing rural settlements is not unique to China. In the Chittagong Hill Tracts of Bangladesh, after the nationalization of forests, with the accelerated reduction of forest area, the intensification of population migration, and the development of settled agriculture, the scale of rural areas has expanded [25]. The development of rural settlements in Argentinian Patagonia is also affected by the local land use policy [26]. In short, national policy is one of the important factors that lead to the current pattern of rural settlements in the Lijiang River Basin. Interestingly, in turn, the spatial pattern landscape of rural settlements is the basis for guiding and formulating land use policies. Therefore, the analysis of the spatial pattern of rural settlements in the Lijiang River Basin has a certain optimization and protection effect on the existing local rural settlement pattern.
Tourism economic development and town distribution are the formation factors of rural settlements in many countries, including China [27], Nepal's Annapurna region [28], Israel [29], and NP Кopaonik (Serbia) [9]. In addition, it has also become a consensus that historical heritage is a vital factor in rural settlements, because in some cases, it may be able to determine the future economic specialization of the region and its prospects for social and economic development [30-33]. Moreover, the influence of minority cultures on the development of rural settlements may be unique to China. Guilin has 28 ethnic minorities (all 55 ethnic groups are minorities except the Han nationality), which is the gathering place of ethnic minorities. Due to the excavation of the Lingqu Canal in history, the Han Chinese took advantage of this waterway to occupy part of the favorable geographical positions to build rural settlements, which are larger and more concentrated. Compared with the Han nationality, due to their fear of external things, ethnic minorities tend to live in deep mountains and valleys with higher elevations and greater slopes. To adapt to the disadvantaged geographical environment, a large number of ethnic ecological wisdom has been generated, such as terraces and stilted buildings, etc., and their rural settlements will be smaller and more scattered due to the geographical conditions.
Sections 4. and 5. could be merged into one integrated section.
Response: Thank you very much for your comments. Although we are more than willing to implement your comments, I really don't understand what sections 4 and 5 are referring to. Perhaps section 4 means “3.4 The distribution of rural settlements in the Lijiang River Basin from the perspective of landscape pattern index” and section 5 means “3.5 Elevation and slope affects the spatial form of rural settlements in the Lijiang River Basin”. However, 3.4 and 3.5 are not suitable for integration into a part. If you still have questions, please feel free to contact us. Thanks again.
We would like to express our great appreciation to you and reviewers for comments on our paper. Looking forward to hearing from you.
Thank you and best regards.
Yours sincerely,
Xianyan Zhang
Reviewer 3 Report
General comments:
- The manuscript does not present scientific relevance; however, it presents an important analysis applied to Lijian River Basin.
- The manuscript fits the aims and scope of the International Journal of Environmental Research and Public Health
- There are some English mistakes that should be improved before acceptance. I recommend an English review by a native speaker.
- In general, the authors do not justify the use of the present methodology, despite they present the methodology. The manuscript fails in the explanatory reasons for such use.
- Highly descriptive instead bring new information. Some results are trivial.
- The authors should present more photos to give an idea of what kinds of the landscape are presented in the basin area. For readers from outside of China can be difficult to understand the landscape.
Specific comments:
- In Figure 1 (line 63), the authors have to present the location of the river basin regarding the entire of China too. For readers from other countries, like me, the information on location is not precise, needing further investigation to understand the region area. In addition, it is a good practice to clearly identify the outlet point in the river basin.
- The authors have used i) land use (30 m resolution); ii) vectorial map with villages location; iii) DEM (30 m resolution); iv) lithologic map
- Despite good arguments concerning the policy decisions and land use in China, the manuscript does not justify appropriately the comments. It is an important discussion that have being made
- Line 298: the manuscript may serve as a theoretical basis for optimizing and protecting rural settlements, nonetheless, the direct use of the manuscript for promoting rural revitalization is not clear in the manuscript.
Author Response
Dear Reviewer 3:
On behalf of my co-authors, we thank you very much for giving us an opportunity to revise our manuscript, we appreciate editor and reviewers very much for their positive and constructive comments and suggestions on our manuscript entitled " Spatial pattern characteristics and causes of rural settlements in Lijiang River Basin based on ArcGIS ". (ID: ijerph-2152077).
We have studied reviewer’s comments carefully and have made revision which marked in red in the paper. Those comments are all valuable and very helpful for revising and improving our paper, as well as the important guiding significance to our researches. We have studied comments carefully and have made correction which we hope meet with approval. The main corrections in the paper and the responds to the reviewer’s comments are as following:
- The manuscript does not present scientific relevance; however, it presents an important analysis applied to Lijian River Basin.
Response: Thanks for the recognition of the significance of this work! We have added some relevant references in the introduction and discussion section in order to improve our manuscript.
- The manuscript fits the aims and scope of the International Journal of Environmental Research and Public Health
Response: Thank you very much for this endorsement.
- There are some English mistakes that should be improved before acceptance. I recommend an English review by a native speaker.
Response: Thank you very much for your suggestions. The revised manuscript has now been carefully proofread and polished by a native English-speaking colleague. We hope that the language of the manuscript has been improved.
- In general, the authors do not justify the use of the present methodology, despite they present the methodology. The manuscript fails in the explanatory reasons for such use.
Response: Thank you very much for your suggestions. We explained the reasons for using each method in the Method section. And the discussion is expanded in the Discussion section. As follows:
In section of “2. Research methods and data sources”
2.3. Getis-Ord General G analyses
Clustering for the density distribution of rural settlements was assessed by using the Getis-Ord General G statistic in ArcGIS v.10.2 software….
2.4. Hot spot analysis (Getis-Ord Gi*)
Hot spot analysis was utilized to explore the spatial cluster arrangements appearing in data. It was performed the Getis-Ord Gi* statistic to identify hot spots (high values) and cold spots (low values) for rural settlements in ArcGIS v.10.2 software, based on the rural settlements of Lijiang River Basin in 2020….
2.5. Kernel density estimation
Kernel density estimation is utilized to predict the unknown density functions in probability theory, which can directly reflect the distribution characteristics of the sample data. The central tendency and dispersion degree of rural settlements was assessed by kernel density estimation…
2.6. Landscape pattern index
Landscape pattern index is a method of describing landscape pattern characteristics using indexed landscape features and a quantitative indicator reflecting the relationship between landscape pattern structure and phenomenological processes. Here, landscape pattern index is applied to the analysis of the relationship of structural features to the spatial distribution in rural settlement landscapes.
In section of “4. Discussion”
Hot spot analysis (Getis-Ord Gi*), kernel density estimation, and landscape pattern index are the main technical tools used in this study to explore the pattern of rural settlements. In fact, they are a common and reliable tool for evaluating the clustering of an element in its spatial location. For example, the spatial and temporal patterns of land use land cover changes and land surface temperature variations in Bengaluru urban district, India, was analyzed using Getis–Ord Gi* statistics [19]. Guerri et al. mapped and evaluated the distribution of thermal summer diurnal hot- and cool-spots associated with greening, urban surfaces, and city morphology in the area of Florence in Tuscany (Italy) using Getis-Ord Gi* spatial statistics [20]. GIS-based kernel density was used to examine the spatial configuration of possible land-use areas through archaeological legacy data to survey landscape dynamics [21] and also applied to estimate the spatial pattern of road density and its impact on landscape fragmentation [22]. Moreover, the landscape pattern index was used to compare land use/land cover change and landscape pattern in two areas in China [23] and has also been utilized to analyze the effects of land use/land cover and landscape patterns on seasonal river water quality in small catchments [24]. In summary, these technical tools are effective strategies for analyzing spatial landscape patterns and spatial configuration characteristics, so they reliably reflect the real spatial pattern of rural settlements in the Lijiang River Basin.
- Highly descriptive instead bring new information. Some results are trivial.
Response: Thank you very much for your comments.
- The authors should present more photos to give an idea of what kinds of the landscape are presented in the basin area. For readers from outside of China can be difficult to understand the landscape.
Response: Thank you very much for your suggestions. We sorted out the village photos collected in our daily work to find photos that can represent the local characteristic landscape. The newly added pictures are as follows:
In the section “3.2. Micro/small-scale villages are mainly located in the upstream, and large villages are mainly located in the middle and downstream”
“…As shown in Supplemental Figure 1, the rural settlements located in the upper section are small and scattered under the influence of the mountains and rivers, and form a typical terraced landscape…”
“…For example, as we can see from Supplemental Figure 2, the rural settlements near Huixian Wetland in the middle reaches have characteristic "glass field" (farmland irrigated, like pieces of glass under the light and shadow), which benefits from the flat terrain of the area, so the rural settlements are large and concentrated…”
Supplemental Figure 3 showed the rural settlement near Xianggong Hill, located in the downstream section.
Moreover, the characteristic houses of the ethnic minorities in Guilin area, "Stilted Building" see Supplemental Figure 4.
Specific comments:
- In Figure 1 (line 63), the authors have to present the location of the river basin regarding the entire of China too. For readers from other countries, like me, the information on location is not precise, needing further investigation to understand the region area. In addition, it is a good practice to clearly identify the outlet point in the river basin.
Response: Thank you for your suggestions. We have presented the location of the river basin regarding the entire of China, and we also clearly identified the outlet point in the river basin. As follows:
- The authors have used i) land use (30 m resolution); ii) vectorial map with villages location; iii) DEM (30 m resolution); iv) lithologic map
Response: Thank you for your comments. Yes, our manuscript contains these works.
- Despite good arguments concerning the policy decisions and land use in China, the manuscript does not justify appropriately the comments. It is an important discussion that have being made
Response: Thank you for your comments. We have done as requested. The discussion has added, as follows:
In China, land in urban areas is owned by the state, while land in rural and suburban areas is owned by collectives unless it is owned by the state as stipulated by law. Individuals only enjoy the right to use and the right to profit from the land. Therefore, national policies can greatly affect the formation of rural settlements. For example, for the "Beautiful Rural Construction" project, the country systematically and comprehensively allocates people, money, and goods to rural areas from six aspects, including village planning, village construction, ecological environment, economic development, public services, and other aspects. This policy has significantly affected the geographical location, population distribution, landscape morphology, and economic development of rural settlements. Therefore, it is obvious that the distribution of rural settlements in the Lijiang River Basin is also affected by national policies. The phenomenon of national policies changing rural settlements is not unique to China. In the Chittagong Hill Tracts of Bangladesh, after the nationalization of forests, with the accelerated reduction of forest area, the intensification of population migration, and the development of settled agriculture, the scale of rural areas has expanded [25]. The development of rural settlements in Argentinian Patagonia is also affected by the local land use policy [26]. In short, national policy is one of the important factors that lead to the current pattern of rural settlements in the Lijiang River Basin. Interestingly, in turn, the spatial pattern landscape of rural settlements is the basis for guiding and formulating land use policies. Therefore, the analysis of the spatial pattern of rural settlements in the Lijiang River Basin has a certain optimization and protection effect on the existing local rural settlement pattern.
- Line 298: the manuscript may serve as a theoretical basis for optimizing and protecting rural settlements, nonetheless, the direct use of the manuscript for promoting rural revitalization is not clear in the manuscript.
Response: Thank you for your comments. The protection role is mainly proposed according to China's rural policy and land use policy. We have added discussion of policy impacts on rural settlements to demonstrate the modest contribution of this study to rural conservation.
In China, land in urban areas is owned by the state, while land in rural and suburban areas is owned by collectives unless it is owned by the state as stipulated by law. Individuals only enjoy the right to use and the right to profit from the land. Therefore, national policies can greatly affect the formation of rural settlements. For example, for the "Beautiful Rural Construction" project, the country systematically and comprehensively allocates people, money, and goods to rural areas from six aspects, including village planning, village construction, ecological environment, economic development, public services, and other aspects. This policy has significantly affected the geographical location, population distribution, landscape morphology, and economic development of rural settlements. Therefore, it is obvious that the distribution of rural settlements in the Lijiang River Basin is also affected by national policies. The phenomenon of national policies changing rural settlements is not unique to China. In the Chittagong Hill Tracts of Bangladesh, after the nationalization of forests, with the accelerated reduction of forest area, the intensification of population migration, and the development of settled agriculture, the scale of rural areas has expanded [25]. The development of rural settlements in Argentinian Patagonia is also affected by the local land use policy [26]. In short, national policy is one of the important factors that lead to the current pattern of rural settlements in the Lijiang River Basin. Interestingly, in turn, the spatial pattern landscape of rural settlements is the basis for guiding and formulating land use policies. Therefore, the analysis of the spatial pattern of rural settlements in the Lijiang River Basin has a certain optimization and protection effect on the existing local rural settlement pattern.
We would like to express our great appreciation to you and reviewers for comments on our paper. Looking forward to hearing from you.
Thank you and best regards.
Yours sincerely,
Xianyan Zhang

Round 2
Reviewer 3 Report
No more comments after the second-round review.